# CXCL13 in Cancer and Other Diseases: Biological Functions, Clinical Significance, and Therapeutic Opportunities

**DOI:** 10.3390/life11121282

**Published:** 2021-11-23

**Authors:** San-Hui Gao, Sheng-Zhi Liu, Gui-Zhen Wang, Guang-Biao Zhou

**Affiliations:** 1State Key Laboratory of Molecular Oncology, National Cancer Center/National Clinical Research Center for Cancer/Cancer Hospital, Chinese Academy of Medical Sciences and Peking Union Medical College, Beijing 100021, China; dlbe2011@foxmail.com (S.-H.G.); gzwang@cicams.ac.cn (G.-Z.W.); 2State Key Laboratory of Membrane Biology, Institute of Zoology, Chinese Academy of Sciences & University of Chinese Academy of Sciences, Beijing 100101, China; 3Department of Pharmacology, School of Pharmaceutical Sciences, Capital Medical University, Beijing 100069, China; szliu@ccmu.edu.cn or; 4Department of Biomedical Engineering, Indiana University-Purdue University Indianapolis, Indianapolis, IN 46202, USA

**Keywords:** C-X-C chemokine ligand 13 (CXCL13), C-X-C chemokine receptor type 5 (CXCR5), cancer, tumor microenvironment

## Abstract

The development of cancer is a multistep and complex process involving interactions between tumor cells and the tumor microenvironment (TME). C-X-C chemokine ligand 13 (CXCL13) and its receptor, CXCR5, make crucial contributions to this process by triggering intracellular signaling cascades in malignant cells and modulating the sophisticated TME in an autocrine or paracrine fashion. The CXCL13/CXCR5 axis has a dominant role in B cell recruitment and tertiary lymphoid structure formation, which activate immune responses against some tumors. In most cancer types, the CXCL13/CXCR5 axis mediates pro-neoplastic immune reactions by recruiting suppressive immune cells into tumor tissues. Tobacco smoke and haze (smohaze) and the carcinogen benzo(a)pyrene induce the secretion of CXCL13 by lung epithelial cells, which contributes to environmental lung carcinogenesis. Interestingly, the knockout of CXCL13 inhibits benzo(a)pyrene-induced lung cancer and azoxymethane/dextran sodium sulfate-induced colorectal cancer in mice. Thus, a better understanding of the context-dependent functions of the CXCL13/CXCR5 axis in tumor tissue and the TME is required to design an efficient immune-based therapy. In this review, we summarize the molecular events and TME alterations caused by CXCL13/CXCR5 and briefly discuss the potentials of agents targeting this axis in different malignant tumors.

## 1. Introduction

Chemokines are a family of chemotactic cytokines with small molecular weights (8–14 kDa) [1]. Chemokines are classified into four groups according to the position of the first two cysteines closest to the amino terminus: C, CC, CXC, and CX3C [2]. Chemokines exert their functions by binding to their receptors, which are seven-transmembrane guanine-protein-coupled receptors (GPCRs) [3]. Chemokines have important roles in regulating lymphoid tissue development, immune homeostasis, and inflammatory responses by directing the migration of leukocytes into the injured or infected tissues [2]. A complex chemokine-chemokine receptor signaling network is critical to the tumor microenvironment (TME), which makes pivotal contributions to tumor cell proliferation, migration, invasion, angiogenesis, and evasion of anti-tumor immunity, facilitating tumor initiation, progression, and metastasis [4,5,6,7,8].

Chemokines and their receptors also modulate lymphocyte populations in the TME, thus inducing resistance to immune checkpoint inhibitors that exhibit remarkable efficacies on a proportion of patients with many cancer types [9,10]. Inconsistent with these observations, targeting chemokine receptors with neutralizing antibodies endow a more sensitized phenotype and enhance responses to immune checkpoint blockades [11,12].

## 2. CXCL13/CXCR5 and Immune Homeostasis

### 2.1. CXCL13/CXCR5: Genes and Proteins

C-X-C chemokine ligand 13 (CXCL13), also known as B-cell attracting chemokine 1 (BCA-1) or B-lymphocyte chemoattractant (BLC), was originally identified as a homeostatic chemokine to attract B cells, a minority of T cells, and macrophages [13]. The human *CXCL13* gene localizes on chromosome 4q21 and encodes CXCL13 protein, which has 109 amino acids, a molecular mass of 12,664 Da, and a crystal structure as below (Figure 1A). The receptor of CXCL13 is the C-X-C chemokine receptor type 5 (CXCR5), which is also named Burkitt’s lymphoma receptor 1 (BLR1) and is defined as a member of the superfamily of seven-transmembrane GPCRs (Figure 1B). CXCR5 has two transcripts, both localized on the cell membrane [14], and is expressed by follicular helper T cells (Tfh) [15], circulating CD4^+^ T cells [16], B cells [17], CD68^+^ macrophages [18], and tumor cells. Moreover, FANCA-mediated CXCR5 neddylation is involved in targeting the receptor to the cell membrane, and CXCR5 neddylation stimulates cell migration and motility [19].

### 2.2. CXCL13/CXCR5 Axis

The precise mechanism of how the CXCR5 receptor responds to CXCL13 and mediates signaling activation has not been fully elucidated. Evidence has demonstrated that CXCR5 interacts with cytosolic and membrane proteins to form heterodimers and heterotrimers, respectively [20,21,22]. CXCR5 couples to cytosolic α, β, and γ subunits of G proteins to form heterotrimeric guanine nucleotide-binding proteins [20]. After CXCL13 binds to CXCR5, G proteins dissociate from CXCR5, dividing into G_α_ and G_βγ_, which stimulate different downstream molecules and subsequently trigger specific intracellular signal transduction pathways [20,23]. The intracellular domains, and probably the transmembrane-spanning domains of CXCR5, are required to activate G proteins [24,25]. CXCR5 can also form heterodimers with membrane proteins, such as CXCR4 and Epstein–Barr virus-induced receptor 2 (EBI2) [20,22]. The EBI2/CXCR5 heterodimer lowers the affinity of CXCL13 for CXCR5 and reduces the activation of G proteins, potentially contributing to the alteration of the CXCR5 binding pocket by heterodimer formation [22].

### 2.3. Physiological Functions of CXCL13/CXCR5

CXCL13 is abundantly expressed on follicular helper T cells (Tfh), follicular dendritic cells (FDCs), and stromal cells in the follicles of secondary lymphoid organs (SLOs) and is essential for the development of the B cell zones of SLOs [13,26,27,28]. SLOs, which include the spleen, lymph nodes, and Peyer’s patches, coordinate antigen-specific primary immune responses via promoting the interactions between antigen-presenting cells and lymphocytes. CXCR5 is expressed by mature B lymphocytes [29], a subpopulation of follicular B helper T cells [30,31], and antigen-bearing dendritic cells (DCs) [32], which control their migration into SLOs towards the gradient of CXCL13 [26,33,34,35]. CXCR5 regulates Burkitt’s lymphoma (BL) lymphomagenesis, B cell differentiation, and migration [25,29,34].

CXCL13 and CXCR5 are required to maintain SLO architecture [26,34,36], whereas deficiencies in CXCL13 [26] and CXCR5 [34] result in the abnormal development of lymph nodes and Peyer’s patches. Consistently blocking the CXCL13/CXCR5 axis inhibits the migration and localization of B cells to lymphoid follicles, which are also called B cell zones in SLOs [13,26,27,34]. CXCL13/CXCR5 signaling enhances B cell receptor (BCR)-triggered B cell activation by shaping cell dynamics [37]. Proline-rich tyrosine kinase (Pyk2) and focal adhesion kinase (FAK) are required for CXCL13-induced chemotaxis of B-2 cells into the lymph follicles and B cells in the marginal zone (MZ), since both the Pyk2 inhibitor and FAK inhibitor suppress CXCL13-induced migration of B-2 cells and MZ B cells (B cells in marginal zone) [38].

In response to CXCL13 secreted by Tfh, FDCs, or marginal reticular cells (MRC), and peripheral CXCR5^+^ B cells are recruited into the lymphoid follicles or germinal center (GC) in the SLOs through high endothelial venules (HEVs) (Figure 2). In the lymphoid follicles, a positive feedback loop mediated by CXCL13 boosts follicle development and sustains SLOs homeostasis. On one hand, CXCL13 secreted by FDCs upregulates membrane lymphotoxin α1β2 (LTαβ) on B cells. On the other hand, LTαβ interacts with the lymphotoxin-β receptor (LTβR) on FDCs and triggers FDC development, mutation, and CXCL13 production (Figure 2). In GCs, the expression of LTαβ on B cells is independent of CXCL13 [26]. In the GCs of Peyer’s patches, the formyl peptide receptor (FPR) expressed on FDCs interacts with its endogenous ligands LL-37, promoting the generation of CXCL13 and B cell activators, and subsequently activates B cells [39]. LL-37-mediated FPR-2 signaling in follicular dendritic cells contributes to B cell activation. The CXCL13/CXCR5 axis also connects Tfh cells with B cells. These CXCR5-expressing Tfh cells migrate to the lymphoid follicles under the chemotaxis of CXCL13, where they subsequently initiate GC formation, BCR affinity maturation, and B cell differentiation into antibody-producing plasma cells and memory cells by providing costimulatory receptors and cytokines (Figure 2) [40,41,42]. Although most B cells in SLOs are B-2 cells, CXCL13 also recruits B-1 cells into the body cavity, which triggers early immune defense by producing the low-affinity antibody IgM [43,44]. Overall, the CXCL13/CXCR5 axis is essential for SLO structure and immunity responses.

## 3. CXCL13/CXCR5 and Non-Cancerous Diseases

CXCL13 signaling is involved in multiple diseases and exhibits context-dependent effects in inflammatory conditions and tumor tissues. Generally, the tertiary lymphoid structure (TLS) will develop in non-lymphoid tissues within or near the pathological sites when an organism suffers from disorders, including persistent infection, autoimmune disease, chronic obstructive pulmonary disease (COPD), and cancer (Figure 3) [45,46]. Structurally, TLSs have B cell zones, T cell zones, GCs, and HEVs.

CXCL13 is aberrantly expressed and acts as the main orchestrator in TLSs [24,45,47,50,51,52,53,54]. After viral infection, type I interferon is produced and can induce CXCL13 production in a population of lung fibroblasts, driving CXCR5-dependent recruitment of B cells and initiating ectopic germinal center formation [55]. Primary pulmonary fibroblast-secreted CXCL13 induces the formation of inducible bronchus-associated lymphoid tissue (iBALT), which drives immune responses to fungal stimulation within the lungs [47]. In one fatal and irreversible interstitial lung disease, idiopathic pulmonary fibrosis, CXCL13 is produced by CD68- and CD206-positive alveolar macrophages, and the serum CXCL13 concentration predicts the progression and severity of the disease [56]. In the lungs of mice and patients bearing COPD induced by chronic cigarette smoke exposure, CXCL13 is elevated in the lymphoid follicles and mediates the formation of TLS, resulting in chronic inflammation in bronchoalveolar lavage and destruction of alveolar walls [45].

In the synovial tissues of rheumatoid arthritis, CXCL13 is produced by PD-1^hi^CXCR5^−^CD4^+^ T cells [57]. CXCL13 is also a key regulator of B cell recruitment to the cerebrospinal fluid in acute Lyme neuroborreliosis [58]. CXCL13 participates in TLS formation in some autoimmune diseases, such as primary Sjögren’s syndrome [59,60], systemic lupus erythematosus, myasthenia gravis [61], and atherosclerosis [62], but not in rheumatoid arthritis [63] or acute Lyme neuroborreliosis [64]. CXCL13 is also highly produced during adipogenesis, and has been shown to be a differentiation- and hypoxia-induced adipocytokine that exacerbates the inflammatory phenotype of adipocytes through the induction of the pleckstrin homology (PH) domain leucine-rich repeat protein Ser/Thr specific phosphatase family 1 (PHLPP1), which regulates AKT activation [65,66].

CXCL13 is upregulated by *Helicobacter suis*, the most prevalent non-*Helicobacter pylori* species colonizing the stomach of humans suffering from gastric disease [67]. CXCL13 is also induced in *Helicobacter*-related chronic gastritis and is involved in the formation of lymphoid follicles and the gastric lymphomas of mucosa-associated lymphoid tissue types [68,69]. In non-alcoholic fatty liver disease (NAFLD), repressed expression of CXCL13 may ameliorate steatosis-related inflammation [70]. CXCL13 drives spinal astrocyte activation and neuropathic pain via CXCR5 [71], and is critical to preserve motor neurons in amyotrophic lateral sclerosis [72]. Further understanding of the regulations and functions of the CXCL13/CXCR5 axis will aid the rational design of therapeutics for these diseases.

## 4. CXCL13/CXCR5 and Cancer

The CXCL13/CXCR5 axis is involved in the regulation of cancer cell survival, apoptosis, proliferation, differentiation, migration, invasion, and adaptive immunity, and shows dichotomic anti- and pro-tumor functions in the TME [21,50,73,74,75,76,77,78] (Figure 4). CXCL13 and CXCR5 have important roles in cancer (Figure 5) and represent potential markers to predict the response to immune checkpoint therapy [51,79,80,81]. These molecules may also serve as novel targets for the development of preventive and/or therapeutic agents for cancer.

### 4.1. CXCL13 Sources within the Tumor and the Tumor Microenvironment

#### 4.1.1. CXCL13: Cellular Sources within TME

CXCL13 is secreted by multiple populations of cells within the TME, including stromal cells, endothelial cells, lymphocytes, and tumor cells. FDC, a considerable stromal cell population, is the major producer of CXCL13 in the GCs [85,86]. Cancer-associated fibroblasts can convert to myofibroblasts and secrete CXCL13 into the TME upon hypoxia and TGF-β stimulation [87]. CXCL13 is also produced by human bone marrow endothelial (HBME) cells [88], Tfh that have infiltrated into tumor tissues [89], PD1^+^ CD8^+^ T cells [90], the TGFβ-dependent CD103^+^CD8^+^ tumor-infiltrating T-cell (TIL) subpopulation [91], neoplastic T cells [85], and several types of tumor cells.

#### 4.1.2. CXCL13: Production under Carcinogen Stimulation

Environmental carcinogens can induce the production of CXCL13. Studies showed that the expression of CXCL13 at both the mRNA and protein levels was increased in B cell areas of lymphoid follicles in the lungs of cigarette smoke (CS)-exposed mice, and the CS-induced upregulation of CXCL13 was confirmed in patients with COPD. Interestingly, CS-induced formation of pulmonary lymphoid follicles was blocked by anti-CXCL13 antibodies in mice, and the absence of tertiary lymphoid organs (TLOs) in bronchoalveolar lavage alleviated the inflammatory response and destruction of the alveolar walls but did not impact the remodeling of the airway wall [45].

CXCL13 plays a critical role in environmental carcinogenesis. Wang et al. [18] screened for abnormal inflammatory factors in patients with non-small cell lung cancers (NSCLCs) from Xuanwei city in China’s Yunnan Province, where the wide use of smoky coal resulted in severe household air pollution, and found that CXCL13 was substantially upregulated in 63 (90%) of 70 Yuanwei patients with NSCLC. In NSCLC patients from control regions where smoky coal was not used, CXCL13 was overexpressed in 44/71 (62%) of smoker patients and 27/60 (45%) of non-smoker patients [18]. Benzo(a)pyrene (BaP), a polycyclic aromatic hydrocarbon (PAH) carcinogen found in tobacco smoking and haze (smohaze) [92], can be metabolically activated by the production of BaP-7,8-diol-9,10-epoxides (BPDEs). BPDE reacts with DNA to form adducts at *N^2^* of deoxyguanosine (BPDE-*N2*-deoxyguanosine), which induces mainly G→T genomic mutations to promote carcinogenesis [93]. We found that BaP induced the production of CXCL13 by lung epithelial cells in vitro and in vivo. Consistent with these observations, CXCL13 was shown to be elevated in serum samples of current and former smokers and was associated with lung cancer risk [94]. CXCL13 induces the production of secreted phosphoprotein 1 (SPP1 or osteopontin) by macrophages to activate β-catenin and induce an epithelial-to-mesenchymal transition (EMT) phenotype (Figure 6). Deficiency in CXCL13 or CXCR5 significantly suppressed BaP-induced lung cancer in mice, indicating that CXCL13 plays a key role in smohaze carcinogen-induced lung cancers [18,92]. CXCL13 is also upregulated in human colorectal cancer and is secreted by dendritic cells [95]. The carcinogen azoxymethane, which is catalyzed into methylazoxymethanol to induce G→A genomic mutations, induces colorectal cancer in vivo. Interestingly, knockout of CXCL13 inhibits azoxymethane/dextran sodium sulfate-induced colorectal cancer in mice [95]. These data suggest a crucial role for the CXCL13-CXCR5 axis in cancers induced by environmental factors and could be a novel target for the development of preventive and therapeutic agents to combat related cancers.

### 4.2. CXCL13/CXCR5 and Cancer Hallmarks

#### 4.2.1. CXCL13 and Cell Proliferation

CXCL13 binds specifically to CXCR5, which couples with MEK/ERK to induce cell proliferation [96]. In clear cell renal cell carcinoma (ccRCC) cells, CXCL13 promotes proliferation by binding to CXCR5 and subsequently activating the PI3K/AKT/mTOR signaling pathway [97]. The PI3K/AKT pathway also plays a key role in the CXCL13/CXCR5 axis, promoting colon cancer growth and invasion [98,99]. The CXCL13/CXCR5 axis promotes the proliferation and invasion of prostate cancer (PCa) cells by activating JNK, ERK, SRC/FAK, PI3K, and Akt [73,96,100]. CXCL13 also promotes the proliferation of androgen-responsive LNCaP PCa cells in a JNK-dependent, DOCK2-independent manner, whereas in androgen-independent PC3 cells, CXCL13-induced proliferation is dependent on DOCK2 [73,100,101]. CXCL13/CXCR5 also promotes cell cycle progression from the G1 to the S phase in PCa cells by the inactivation of CDKN1B and the activation of Cdk2 [73]. In addition, CXCL13 is involved in the progression of breast cancer cells through the CXCR5/ERK pathway [102]. Therefore, the CXCL13/CXCR5 axis plays a key role in regulating the proliferation of many cancer cells and could be a valuable therapeutic target.

#### 4.2.2. CXCL13 and Cell Apoptosis

The CXCL13/CXCR5 axis plays an important role in cell homeostasis, as well as helping leukemic cells escape apoptosis by regulating chemokine-induced signaling [103]. In breast cancer cells, the decrease in CXCL13 leads to the decreased expression of CXCR5, p-ERK/ERK, and cyclin D1 as well as the increased expression of cleaved Casp-9, which is an initiator caspase protease for apoptosis [104]. Moreover, CXCL13/CXCR5 has been demonstrated to induce significant resistance to TNF-α-mediated apoptosis in B cell lineage acute and chronic lymphocytic leukemia (B-ALL and B-CLL) [105,106]. CXCL13 regulates the phosphorylation of Bcl-2 (at Serine 70), Bcl-xL (at Serine 62), and BAD (at Serine 112 and 136) in PC3 cells to exert anti-apoptotic effects [73]. CXCR5 may be involved in the protection of retinal pigment epithelium (RPE) and retinal cells from aging-related photoreceptor apoptosis [107]. These data demonstrate that the CXCL13/CXCR5 axis can confer the evasion of apoptosis in cancer cells by modulating p-ERK/ERK, TNF-α, Casp-9, and other signal pathways.

#### 4.2.3. CXCL13 and Cancer Stem Cell (CSC)

Cancer stem cell (CSC) is a type of tumor cell with the abilities of self-renewal, differentiation, and high drug resistance [108]. IL30 overproduction by prostate cancer stem cell-like cells promotes tumor initiation and development, which involves increased proliferation, vascularization, and myeloid cell recruitment. Moreover, it promotes stem cell-like cell dissemination to lymph nodes and bone marrow by upregulating the CXCR5/CXCL13 axis [109]. CXCL13 recruits B cells to prostate tumors to promote castrate-resistant cancer progression by producing lymphotoxin, which activates an IκB kinase α (IKKα)-BMI1 module in prostate cancer stem cells [110]. The role of the CXCL13/CXCR5 pathway in the cancer stem cells of other malignancies remains to be investigated.

#### 4.2.4. CXCL13 and Drug Resistance

CXCL13/CXCR5 plays an essential role in drug resistance. In multiple myeloma (MM), CXCL13 secreted by mesenchymal stem cells (MSCs) confers resistance to bortezomib to MM cells [111]. In 5-fluorouracil (5-Fu)-resistant colorectal cancer patients, serum CXCL13 is elevated, and a high CXCL13 concentration is associated with a worse clinical outcome [112]. CXCL13 is significantly increased in diffuse large B-cell lymphoma resistance to chemotherapy and is involved in tumor progression [113]. CXCR5 is overexpressed in mantle cell lymphoma (MCL), where it mediates MCL-stromal cell adhesion and drug resistance. The drug resistance of MCL is associated with increased expression of B-cell activation factor (BAFF), which induces the expression of CXCL13 [114].

#### 4.2.5. CXCL13/CXCR5 in the Tumor Microenvironment

The CXCL13/CXCR5 axis may have different roles in the TME. In leukemia, prostate, lung, pancreatic, colon, and gastric cancers, CXCL13 exhibits pro-cancer effects by recruiting B cells [86,87,115,116], CD68^+^ macrophages [18], regulatory B cells (Bregs) [117,118], Treg [119], and CD40^+^ MDSCs [120], shaping an immune-suppressive TME to trigger tumorigenesis and tumor progression (Figure 7A). A few reports regarding breast and lung cancers have shown that the CXC13/CXCR5 axis attracts B cells and Tfh [89,90] to shape the TLS in the peritumoral or tumor sites (Figure 7B), which is associated with adaptive anti-tumor humoral responses and predicting responses to PD-1 blockade therapy.

Increasing evidence indicates that the CXCL13/CXCR5 axis influences lymphocyte infiltration in the TME by regulating cell interactions [96]. CXCL13 plays a key role in the microenvironment of diffuse large B-cell lymphoma (DLBCL) [113]. In NSCLC, intratumoral CD8^+^ T lymphocyte populations with a high level of PD-1 (PD-1^T^) express higher levels of CXCL13 and secrete more CXCL13 than CD8^+^ T cells with intermediate (PD-1^N^) and no PD-1 expression (PD-1^−^). These PD-1^T^ tumor-infiltrating lymphocytes play an active role in the recruitment of immune subsets to the TME via the secretion of CXCL13 and show predictive potential for response to PD-1 blockades [90]. NSCLC patients also have higher levels of serum CXCL13 as compared to healthy controls. Upon CXCL13 stimulation, NSCLC cells with a high level of CXCR5 expression exhibit a pro-migratory phenotype [96]. In prostate cancer, CXCL13/CXCR5 interactions promote the progression of tumor cells, including proliferation and metastasis, which are triggered by multiple signaling cascades, such as ERK, PI3K/Akt, stress-activated protein kinase (SAPK)/c-Jun kinase (JNK), Rac, and protein kinase C epsilon (PKCε)/nuclear factor kappa B (NF-κB) [100,101,122,123]. Intratumoral CXCL13^+^ CD8^+^ T cells orchestrate immune-evasive actions, which consist of increased regulated Tregs and exhausted cytotoxic T cells in gastric cancer [124]. These intratumoral CXCL13^+^ CD8^+^ T cells are associated with poor clinical outcomes and a decreasing response to chemotherapy. CXCL13/CXCR5-mediated recruitment of CD40^+^ myeloid-derived suppressor cells (MDSCs) might induce the immune escape of gastric tumors through inhibiting recruitment of T cells in the TME [120]. Recently, Cabrita et al. [51] found that the coexistence of tumor-associated CD8^+^ T cells and CD20^+^ B cells improved survival in patients with metastatic melanomas; immunofluorescence staining of CXCR5 and CXCL13 in combination with CD20 showed the formation of TLSs in these CD8^+^ CD20^+^ tumors.

#### 4.2.6. CXCL13 and Angiogenesis

Angiogenesis is a distinguishable characteristic of successful tumor growth in all solid tumors, and CXC chemokines are pleiotropic in their ability to regulate tumor-associated angiogenesis, as well as cancer cell metastases [125]. Chronic hypoxia increases the expression of CXCL13 in adipocytes [65] and promotes the metastasis of prostate cancer by increasing the expression of CXCL13 in tumor myofibroblasts [87]. Fibroblast growth factor-2 (FGF2) is a member of the family of the heparin-binding FGF growth factors with pro-angiogenic activity. CXCL13 inhibits FGF2-induced chemotaxis and proliferation, as well as the survival of endothelial cells, acting as an angiostatic chemokine [126]. CXCL13/CXCR5 axis also facilitates angiogenesis during rheumatoid arthritis progression [127].

#### 4.2.7. CXCL13 and Immunometabolic Responses

An integrated immunometabolic response during negative energy balance is required for host survival, and the impacts of nutritional status on immune responses remain to be determined. Recent studies have shown that temporary fasting significantly reduces the number of lymphocytes in Peyer’s patches, whose cellular composition is conspicuously altered after resuming feeding, with the numbers seemingly restored. In this process, nutritional signals are necessary to maintain CXCL13 expression by stromal cells [128]. Fasting reduces the numbers of circulating monocytes, as well as monocyte metabolic and inflammatory activity, while hepatic energy-sensing regulates homeostatic monocyte numbers via CCL2 production [129]. However, the potential roles CXCL13 plays in cancer metabolism remain to be investigated.

#### 4.2.8. CXCL13 and Cancer Metastasis

More than 90% of cancer deaths are attributed to metastasis. The intricate interactions of a chemokine and its receptor play an essential role in tumor metastasis. CXCL13/CXCR5 also participates in the metastasis of multiple cancers. CXCL13 enhances cancer metastasis signaling in an autocrine or paracrine manner, since it is secreted by tumor cells or other cell types, such as stromal cells and lymphocytes. In a murine prostate cancer model, which exhibits PKCε overexpression and *Pten* deficiency, the release of CXCL13 by tumor cells was upregulated in a non-canonical NF-κB pathway, boosting tumor cells’ migratory properties [122]. CXCL13 facilitates breast cancer cell line migratory activity via the nuclear factor kappa-B ligand (RANKL)-Src pathway, which mediates the upregulation of EMT regulators and matrix metalloproteinase-9 (MMP9) [83,130]. CXCL13, also secreted by stromal cells, upregulates the expression of RANKL on stromal cells, promoting tumor cell migration and lymph node metastasis via the RANK-RANKL pathway [131]. CXCL13 mediates distal metastasis of colon cancer by increasing the secretion of MMP13 and the activation of the PI3K/Akt pathway [132]. A further study showed that polarized M2 macrophages mediate premetastatic niche formation and facilitate colorectal cancer liver metastasis by forming a positive-feedback loop of CXCL13/CXCR5/NFκB/p65/miR-934 [133]. On the other hand, CXCL13 recruits CXCR5^+^ CD68^+^ macrophages secreting SPP1, which triggers tumor cell migration through the EMT pathway in lung cancer [18]. The endpoint of facilitating metastasis may arouse enthusiasm in pursuing CXCL13/CXCR5 as a potential target for cancer therapy.

### 4.3. Regulation of CXCL13 in Tumors

A series of studies have shed new light on the regulation of CXCL13 and CXCR5 in tumors. RelA, a subunit of the NF-κB family [134], directly binds to the CXCL13 promoter and positively regulates the transcription of CXCL13, while nuclear factor erythroid 2-related factor 2 (NRF2) acts as a negative transcriptional regulator of this chemokine [135]. CXCR5 was positively regulated by RelA and negatively by p53 [135,136], and nuclear raf-1 kinase regulates the CXCR5 promoter by associating with NFATc3 [137]. P53 homologues, p63 and p73 [138], utilize the same mechanism by which the activity of NFκB is attenuated to reduce the expression of CXCR5 [139]. The aryl hydrocarbon receptor (AhR), a ligand-activated transcription factor, is translocated to the nucleus under BaP stimulation and binds to the xenobiotic-responsive element (XRE) in the promoter of *CXCL13*, positively regulating the transcription of CXCL13 (Figure 6) [18]. Another transcription regulator, interferon regulatory factor 5 (IRF5), directly targets CXCL13 by binding to its promoter and upregulating CXCL13 expression [17]. *CXCL13* is also identified as a downstream target gene of the transcription factor androgen receptor (AR) [140]. In addition, oncoprotein PKCε overexpression with tumor suppressor *Pten* deficiency boosts the expression of CXCL13 individually and synergistically through the non-canonical NF-κB pathway [122]. In a murine *Kras^G12D^ Hif1α* knockout model with *LSL*-*Kras*^+/G12D^ and *Pdx1*-cre with interleukin-1β (IL-1β) overexpression, the increase in CXCL13 levels depends on the combination of HIF1α and Kras, as well as the cooperation of IL-1β and Kras [117,118]. However, the mechanisms of CXCL13 upregulation in both murine models have not been elucidated.

## 5. CXCL13/CXCR5 in Several Cancer Types

Mounting evidence demonstrates a high concentration of CXCL13 and/or high expression of CXCR5 in tumor tissues or tumor cell lines. The CXCL13/CXCR5 axis in both hematological malignancies and solid tumors mediates multiple intracellular signal cascade reactions and yields various phenotypes responding to the signaling pathways. In addition, the CXCL13/CXCR5 axis also potentiates the crosstalk between tumor cells and lymphocytes or non-lymphocytes, shaping a complex TME. The roles of the CXCL13/CXCR5 axis participating in the malignant tumors are context-dependent, including pro-tumor and anti-tumor activities (Figure 7). On one hand, CXCL13 attracts immunosuppressive cells to mediate immune suppression or evasion, leading to tumor progression, while on the other hand, the CXCL13/CXCR5 axis elicits tumoricidal immunity signaling to escape tumor immunosurveillance in some cancer types [54,89,141] (Table 1).

### 5.1. Chronic Lymphocytic Leukemia and Lymphoid Neoplasms

#### 5.1.1. Chronic Lymphocytic Leukemia

Chronic lymphocytic leukemia (CLL) is the most frequently diagnosed subtype of leukemia in adults and is a lymphoproliferative disorder that is characterized by the expansion of monoclonal, mature CD5^+^ CD23^+^ B cells in the peripheral blood, secondary lymphoid tissues, and bone marrow. Most CLL tumor cells are inert and arrested in the G0/G1 of the cell cycle, and there is only a small proliferative compartment; however, the progressive accumulation of malignant cells will ultimately lead to symptomatic disease [142,143]. CLL patients express high levels of membrane CXCR5 on leukemic cells and have an elevated serum concentration of CXCL13 when compared with healthy donors [142,143]. In the murine *Eμ-Tcl1* model of CLL, the CXCR5-expressing malignant B cells are trafficked to GC light zones of B cell follicles in SLOs, where leukemic B cells contact FDCs [86]. Leukemic B cell-associated LTαβ activates LTβR on the FDCs [26,144], which maintains FDC development and structure. Reciprocally, FDCs secrete CXCL13, leading to the potentiation of the CXCL13/CXCR5 axis-mediated malignant B cell attraction and proliferation of leukemic B cells [86]. This crosstalk is repressed by the inhibition of the LTαβ-LTβR or CXCL13/CXCR5 signaling cascades, resulting in attenuation of CLL progression [86]. In addition to potentiating tumor-stromal cell crosstalk, the CXCL13/CXCR5 axis exerts intracellular signaling cascade reactions, which are conducive to malignant cell survival and resistance to apoptosis. When stimulated with CXCL13, CLL B cells commit to actin polymerization, CXCR5 endocytosis, and the activation of p44/p42 mitogen-activated protein kinase (MAPK) [143]. CXCL13, as well as other homeostatic chemokines, including CXCL12, CCL21, and CCL19, make crucial contributions to B-CLL cell survival through phosphorylating MAPK extracellular signal-regulated kinase (ERK)1/2, p90RSK, and protein kinase B (Akt), and inhibiting phosphorylation of the downstream effectors GSK3α/β and FOXO3a [103]. The malignant B cells from CLL or acute lymphocytic leukemia (ALL) patients take advantage of the CXCR5/CXCL13 axis and the CCR7/CCL19 axis for resistance to TNFα-induced apoptosis via upregulation of paternally expressed gene 10 (PEG10) and subsequent stabilization of caspase-3 and caspase-8 [105]. Taken together, the CXCL13/CXCR5 axis assists B-CLL cell migration, localization, survival, and expansion in the microenvironment of SLOs.

#### 5.1.2. Lymphoid Neoplasms

Lymphoid neoplasms, characterized by the malignant clonal proliferation of lymphocytes, such as mature B cells, T cells, and natural killer (NK) cells, are classified as Hodgkin’s lymphoma (HL) and non-Hodgkin’s lymphoma (NHL), according to morphologic and immunologic characteristics [145,146]. Most lymphoid neoplasms are not remediable and eventually relapse after conventional treatment, a phenomenon in which the interaction between malignant lymphocytic cells and resident stromal cells exert a dominant influence [142]. The CXCL13/CXCR5 axis plays pivotal roles in the dissemination and accumulation of malignant lymphocytic cells and in shaping such a tumor-stromal cell microenvironment interaction network [147,148,149]. The expression level of CXCL13 and CXCR5 are dramatically elevated in NHL [150,151,152,153,154]. A plethora of functional evidence has demonstrated that CXCL13 and/or CXCR5 participate in the pathogenesis of lymphoma, including mantle cell lymphoma (MCL) [155], follicular lymphoma (FL) [156], diffuse large B-cell lymphoma (DLBCL) [156], primary intraocular lymphoma (PIOL) [157], primary central nervous system lymphoma (PCNSL) [158,159,160], extranodal natural killer (NK)/T-cell lymphoma (ENKTL) [161], and angioimmunoblastic T-cell lymphoma (AITL) [162,163]. Not only do the CXCL13^+^ and/or CXCR5^+^ malignant lymphocytes (B cells and Tfh cells) promote the accumulation and proliferation of lymphoma cells, but so do the CXCL13^+^ FDC and circulating CXCR5^+^ CD4^+^ T cells [16,85]. CXCL13 is also involved in lymphoproliferative disorders and lymphoid follicular formation in cutaneous B-cell lymphoma (CBCL) [164,165] and *H. pylori*-induced gastric mucosa-associated lymphoid tissue (MALT) lymphomas [68,69]. Due to the function of CXCL13 in lymphoma, mounting studies have demonstrated that CXCL13 serves as a useful marker for the diagnosis and prognosis of PCNSL [158], AITL [162,163], MCL [155], PIOL [157], and ENKTL [161].

### 5.2. Lung Cancer

Chronic inflammation provides a favorable context for lung carcinogenesis [166]. CXCL13 has a higher concentration in the serum of NSCLC patients who have had a history of smoking, as compared with those with COPD [167]. The elevated level of CXCL13 in the serum serves as a risk factor for the progression of the early stages of lung adenocarcinoma [94,168]. In NSCLC cells, a study showed that the expression of CXCR5 in the nucleus is higher than that in the cell membrane [169]. CXCL13 could be used to determine the origin of squamous cell lung cancer in patients with head and neck squamous cell carcinoma (HNSCC), where a risk for lung metastasis exists [170]. Consistently, NCI-H1915 cells with a higher level of CXCR5 expression show more potential to migrate in response to CXCL13 compared to SW-1271 cells with a lower level of CXCR5 [169]. The CXCL13/CXCR5 axis is associated with polycyclic aromatic hydrocarbon(PAH)-induced lung carcinogenesis [18]. Both normal lung epithelial (16HBE) and lung adenocarcinoma (A549) cells show upregulated CXCL13 in a dose- and time-dependent manner when the cells are treated with BaP [18]. Moreover, in the NOD/SCID mice bearing A549-Luc-CXCL13 cells, CXCL13 recruits CXCR5^+^ CD68^+^ macrophages to the TME, where macrophages produce secreted phosphoprotein 1 (SPP1) to promote cell migration and tumor progression. Interestingly, deficiency in CXCL13 or CXCR5 significantly inhibits BaP-induced lung cancer in mice, indicating the critical role of this axis in environmental lung carcinogenesis [18].

Anti-tumor activity of the CXCL13/CXCR5 axis has also been reported, which is shown to be able to promote the formation of TLSs and is linked to improved survival of lung cancer patients [15,54,171]. CXCL13, produced by PD1^+^ CD8^+^ T cells, mediates the immune cells recruited to TLSs. As a result, the TLS is infiltrated by CXCR5^+^ CD4^+^ T cells, CD4^+^ Bcl6^+^ Tfh cells, and B cells [90]. Another study reported that the level of IL-21, IL-10, and CXCL13 are upregulated in PD-1^+^ CXCR5^+^ CD4 T cells in the GCs, which promotes CD8^+^ T cells to produce IFN-γ, induces the proliferation of B cells, and potentiates B cells to produce IgM and IgG, forming an anti-tumor immunity microenvironment [15]. Furthermore, the overexpression of CXCR5 on intratumoral natural killer (NK) cells has been speculated to be involved in their migration in the tumor site, where NK cells participate in tumor immunosurveillance [121]. Although CXCL13 mediates the recruitment of immune cells to the TLS in the intra- and extra-tumor regions, the specific functions and molecular mechanisms of these cells still need to be further investigated.

### 5.3. Prostate Cancer

Prostate cancer (PC) represents a leading cause of cancer-related mortality among men due to its metastasis [172]. Chemokines and homologous receptors play a crucial role in the initiation and progression of metastasis in PC. Extensive studies have reported that the CXCL13/CXCR5 axis mediates PC cell migration, invasion, cell adhesion, and anti-apoptosis functions through regulating the intracellular signaling networks in an autocrine or paracrine fashion and forming an oncogenic microenvironment, which consists of multiple cell types, such as inflammatory cells and PC cells. In PC tissues and cell lines, the expression level of CXCR5 is notably increased and positively related to the progression of PC [173]. CXCR5 mediates PC-cell survival and metastasis under CXCL13 stimulation by coupling with specific G protein subunits [20]. CXCL13, highly expressed in PC tissues, is upregulated by AR and mediates PC genesis and development [140]. PKCε overexpression with a *Pten* deficiency upregulates the release of CXCL13, leading to tumorigenesis and metastasis of PC through the CXCL13/CXCR5 axis in an autocrine manner [122]. Based on antibody microarray analysis, the molecular mechanisms of the CXCL13/CXCR5 axis participating in PC cell proliferation and motility mainly include the PI3K/Akt/cyclin-dependent kinases (Cdk)1/2-Cdk inhibitor 1B (CDKN1B), stress-activated protein kinase (SAPK)/c-Jun kinase (JNK), Src/Erk1/2, and integrin β3-FAK/Src-Paxillin (PXN) pathways [73,100,101,140]. However, the potential of drug targets exploiting these mechanisms to better treat PC require further validation.

CXCL13 also has an essential role in configuring a complex TME. Although patients initially benefit from the treatment of androgen ablation, the responses are transient with a duration of 12–18 months when they progress to castration-resistant (CR) PC [174,175,176]. Under androgen ablation therapy, regressing PC is accompanied by cell death and hypoxia, which activates cancer-associated myofibroblasts (CAMF) to secrete CXCL13 by inducing HIF-1 activation and TGF-β expression [87]. CXCL13 expressed by myofibroblasts mediates B-cell recruitment into the TME [115].In J_H_^−/−^mice lacking mature B cells, the emergence and expansion of castration resistant prostate cancer (CRPC) after androgen ablation relies on B cells, rather than T cells, and the CRPC microenvironment is subsequently infiltrated by B cells [115]. The inflammation-responsive IκB kinase (IKK) β in B cells mediates LTαβ secretion [115]. The interaction between LTαβ and LTβR on PC cells triggers the IKKa-E2F1-BMI1 pathway, which regulates the regeneration of oncogenic prostate stem cells [177,178] and phosphorylates STAT3, an anti-apoptosis and oncogenic transcription factor [179], as a result of the reconstruction of PC cells, accelerating the emergence of CRPC [87,115]. Apart from the recruitment of tumor-infiltrating B cells, CXCL13, expressed by interleukin-6 (IL-6)-treated human bone marrow endothelial (HBME) cells, mediates PC-cell invasion, adhesion to HBME cells, and α_v_β_3_-integrin clustering [88].

### 5.4. Breast Cancer

CXCL13 was found to be overexpressed in breast cancer and in the peripheral blood of patients with breast cancer [180,181]. The CXCL13/CXCR5 axis shows anti-tumor and pro-tumor functions. One study showed that the CXCL13/CXCR5 axis was a good prognostic marker for breast cancer [17,180]. The CXCL13/CXCR5 axis is related to improved outcomes of human epidermal growth factor receptor 2 (HER2)-positive breast cancer [182]. Another study demonstrated that CXCL13, which is regulated by interferon regulatory factor 5 (IRF5), mediates CXCR5^+^ B cells and T cells homing to tumors, thereby eliciting an anti-tumor immune response [17,183]. Infiltrated CD4^+^ Tfh cells express CXCL13, which has been speculated to initiate GC formation and enhance TLS development by homing immune cells to peritumoral or tumor sites [89,184]. In addition, these CXCL13-producing Tfh cells reverse Treg-mediated immune suppression and present adaptive anti-tumor humoral responses [89].

However, a large number of studies have demonstrated that the CXCL13/CXCR5 axis is closely associated with breast cancer growth and lymph node metastasis [83,102,136,185]. Breast cancer cell lines undergoing CXCL13 stimulation overexpress EMT regulators and MMP9 via the nuclear factor kappa-B ligand (RANKL)-Src pathway, which is critical for breast cancer cell progression and migration [83,130]. Breast cancer cells expressing CCL21 also recruit RORγt^+^ innate lymphoid cell group 3 (ILC3) to the TME, where ILC3 stimulates stromal cells to secrete CXCL13 [131]. CXCL13, in turn, potentiates the interaction between ILC3 and stromal cells, resulting in the increased production of RANKL on stromal cells, which promotes tumor cell migration and lymph node metastasis by the RANK-RANKL pathway [131]. After treatment with an anti-CXCL13 antibody, the levels of transforming growth factor beta-1 (TGF-β1), interleukin-1 (IL-1), tumor necrosis factor (TNF), p-Erk/Erk, and cyclin D1 in MDA-MB-231 cells were obviously decreased, whereas cleaved caspase-9 was increased, leading to growth inhibition and apoptosis induction of MDA-MB-231 cells [104]. Moreover, in a female BALB/c mouse model of breast cancer, an anti-CXCL13 antibody reduced tumor growth by impairing the CXCR5/Erk pathway and inducing tumor cell apoptosis [102]. A recent study showed that CXCL13 expression is increased in the sera of breast cancer patients with brain metastases [186], though the role and mechanism of action of the CXCL13/CXCR5 axis in cancer metastasis remain to be elucidated.

### 5.5. Pancreatic Cancer

In pancreatic cancer, CXCL13 signaling participates in the establishment of the pro-oncogenic microenvironment by recruiting tumor-associated B cells [116,117,118]. The murine *Kras^G12D^Hif1α^KO^* model of pancreatic cancer exhibits an abundant expression of CCL19, CCL20, CCL21, CXCL12, and CXCL13, all of which mediate Breg migration and infiltration into tumor tissue to promote pancreatic tumorigenesis [118]. In *Kras^G12D^*-harboring pancreatic ductal adenocarcinoma (PDAC), CXCL13 expression by stromal fibroblasts enhances the infiltration of IL-35 producing B cells into the TME, which potentiates the expansion of pancreatic cancer cells [116]. Recently, the pro-tumor role of B cells in the PDAC has been further complemented. In the PDAC mouse model of *LSL*-*Kras^+/G12D^* and *Pdx1*-cre with IL-1β overexpression, the combination of IL-1β stimulation with *Kras* mutation mediates the upregulation of CXCL13 expression and the expansion of Breg and PD-L1^+^ B cells, which play an immune-suppressive role in inhibiting CD8^+^ T cell activity, in addition to facilitating pancreatic tumorigenesis [117]. These studies indicated that the concentration of CXCL13 might serve as a potential marker for B-cell amplification in PDAC, and immunotherapy that targets CXCL13 signaling, or B cells, might enhance the therapeutic efficacy.

### 5.6. Colorectal Cancer

In colorectal or colon cancer, the CXCL13/CXCR5 axis mediates the pathogenesis, development, and distal metastasis of tumor cells through upregulating the expression and secretion of MMP13, as well as activating the PI3K/Akt pathway [98,125,187,188]. The CXCL13/CXCR5 axis is also involved in the colorectal TME. The CXCL13/CXCR5 axis in histidine decarboxylase (HDC^+^) granulocytic myeloid cells promotes Foxp3 expression, Stat3 phosphorylation, and Treg proliferation, as a result of immune suppression and tumorigenesis [119]. CXCL13 is critical to colorectal cancer pathogenesis, because CXCL13 deficiency and the blockade of CXCL13 signaling ameliorates disease progression. CXCL13 promotes intestinal tumorigenesis through the activation of the AKT signaling pathway in a CXCR5-dependent manner. Translocation of the intestinal microbiota drives CXCL13 production in DCs through the activation of NF-κB signaling, and inhibition of microbiota translocation decreases CXCL13 production and suppresses intestinal tumorigenesis. The carcinogen azoxymethane/dextran sodium sulfate induces colorectal cancer in mice, whereas knockout of CXCL13 significantly inhibits the induction of colorectal tumorigenesis in vivo by these carcinogens [95]. These results show that the CXCL13-CXCR5 axis is involved in the crosstalk between chemokines and cell growth during the development of colorectal carcinogenesis, which provides a therapeutic strategy for targeting CXCL13/CXCR5 in the future clinical treatment of colorectal cancer.

### 5.7. Oral Squamous Cell Carcinoma

The CXCL13/CXCR5 axis makes prominent contributions to oral squamous cell carcinoma (OSCC) invasion of bone/osteolysis [74,84,189]. Overexpression of CXCL13 in the stromal/preosteoblast cells significantly increases the phosphorylation of c-Myc and c-Jun and upregulates the transcriptional regulator NFATc3 [74,84]. Phosphorylated c-Myc (p-C-Myc), p-c-Jun, and NFATc3 bind to the RANKL promoter region and upregulate the expression of RANKL, which participates in osteoclastogenesis and OSCC invasion of the bone [74,84]. Inversely, oral cancer-associated TLSs with upregulation of CXCL13 are linked to the extended survival of oral cancer patients, indicating the roles of TLSs as a prognostic marker and immune treatment for oral cancer [190].

### 5.8. CXCL13 and Other Cancers

CXCL13 abounds in the serum and tumor tissue of patients with gastric cancer and serves as a prognostic marker for patients under postoperative adjuvant chemotherapy [14,76,191,192,193]. The CXCL13/CXCR5 axis drives CD40^+^ MDSC migration to gastric cancer, leading to immune escape and tumor progression [120]. CXCL13^+^ CD8^+^ T cells mediate immune evasion with increased Treg cells and exhausted cytotoxic T cells [124]. High infiltration of CXCL13^+^ CD8^+^ T cells in tumor tissue is associated with poor clinical outcomes for the patients, and elimination of these cells could be helpful for gastric cancer immunotherapy. CXCL13 has been reported to be increased in the serum of patients with hepatocellular carcinoma (HCC) [194,195]. CXCL13 potentiates the progression of HCC by activating the Wnt/β-catenin pathway and promoting IL-12, IL-17, and IgG4 production [194]. The CXCL13/CXCR5 axis has also been reported in the initiation and progression of other solid tumors, such as renal cell carcinoma, neuroblastoma, thyroid cancer, osteosarcoma, ovarian cancer, and melanoma. Taken together, CXCL13 and CXCR5 usually act as oncogenic cascades in the promotion of tumor cell survival, proliferation, migration, and invasion, and represent therapeutic targets for the development of novel anti-cancer drugs.

## 6. Therapeutic Potentials of CXCL13/CXCR5 Axis in Cancer

### 6.1. Therapeutic Effect of Cancer Cells in Targeting CXCL13/CXCR5 or the Downstream Molecules

The CXCL13/CXCR5 axis makes pivotal contributions to the initiation and progression of tumors, and therefore may serve as an attractive therapeutic target for related malignant neoplasms [20,104,112]. Pivotal molecules of the CXCL13/CXCR5 axis may also serve as targets for drug development (Table 1). Intriguingly, utilizing intrakine CXCL13-KDEL, which traps CXCR5 in the endoplasmic reticulum, causes a prolonged growth arrest of cancer cells [99]. In breast cancer, an anti-CXCL13 antibody suppressed tumor growth by inhibiting the TGF-β1, IL-1, TNF, cyclin D1, and CXCR5/Erk pathways [81,86] and repressed tumor metastasis by inhibiting RANKL production [117]. Anti-CXCL13 antibodies and RNAi were used to treat lung cancer and PDAC [18,98]. Small interference RNA (siRNA or shRNA)-mediated silencing of this axis reversed 5-fluorouracil-resistance in colorectal cancer cells or restrained the volume and mass of tumors in murine models [112,122]. A study showed that spebrutinib (CC-292), a small molecular inhibitor of Bruton’s tyrosine kinase (BTK), significantly reduces the serum concentration of CXCL13 and shows a therapeutic effect on rheumatoid arthritis [196]. Inhibition of the downstream molecules of the CXCL13/CXCR5 axis, e.g., TGFβR and NFATc3, represent an alternative therapeutic strategy for cancers [103,115,131]

### 6.2. Regulating the Non-Cancerous Cells in the TME by Directly or Indirectly Targeting the CXCL13/CXCR5 Axis

CXCL13-producing non-cancerous cells in the TME, such as myofibroblasts, myeloid cells, B cells, and T cells, may also serve as potential targets for cancer therapy (Table 1). A previous study demonstrated that blocking the CXCL13/CXCR5 axis using siRNA dampens the recruitment and expansion of Tregs, which exert immunosuppressive effects in the TME [119]. B cells homing into the TME via the CXCL13/CXCR5 axis include pro-tumor B cells and anti-tumor B cells. In pro-neoplastic circumstances, tumor-infiltrating B cells facilitate tumor progression by producing cytokines to enhance cancer cell survival and proliferation [115,116], or by developing immune-suppressive B cells to impair CD8^+^ T cell activity [117]. Therapeutic strategies (including anti-CXCL13 antibody) to block B cell recruitment have efficiently hindered cancer progression and prevented cancer-associated myofibroblasts (CAMF) activation by phosphodiesterase 5 (PDE5) inhibitors or by deleting CAMF along with blocking the TGF-β receptor kinase [87]. In anti-neoplastic circumstances, B cells engage in responses against tumors by secreting immunoglobulins, activating T cells, and directly lysing cancer cells [15,49]. Emerging evidence has depicted the upregulation of CXCL13 levels in the TLS, where B cells yield the capability of releasing antibodies and presenting antigens to CD8^+^ T cells [190,191]. An increased interest in developing a responsive biomarker and target for immune checkpoint therapy has focused on the role of B cells and the TLS, which are associated with favorable outcomes for patients after immunotherapy [51,141,195,197]. Given the crucial role of CXCL13/CXCR5 signaling in B cell migration and TLS formation, a further understanding of how these processes are regulated by the CXCL13/CXCR5 axis might provide a novel strategy to enhance the response to immune checkpoint blockade therapy.

## 7. Concluding Remarks

The CXCL13/CXCR5 axis plays multifaceted roles in intracellular signaling transduction pathways and interactions among tumor cells, stromal cells, and lymphocytes. The CXCL13/CXCR5 axis not only modulates molecular events inside malignant cells to promote tumor initiation and progression, but also recruits multiple populations of lymphocytes to exert pro-tumor or anti-tumor immunity reactions in the TME. An intriguing phenomenon is that those anti-neoplastic lymphocytes attracted by CXCL13 signaling, such as B cells and Tfh, participate in the organization of the TLS. However, the circumstances, which contain immune-suppressive cells recruited by the CXCL13/CXCR5 axis, with or without TLSs, have not been fully illuminated. In addition, CXCL13-expressing CD8^+^ T cells are linked to the proinflammatory features of macrophages and show enhanced cytotoxicity following anti-PD-L1 therapy in triple-negative breast cancer [198]. Thus, better recognition of the specific microstructures in the TME might allow for the development of optimal treatment strategies. Additionally, understanding whether CXCL13/CXCR5 signaling could be an untapped target for inducing the recruitment of B cells and the formation of TLSs in the TME could complement current immunotherapy.

## Figures and Tables

**Figure 1 life-11-01282-f001:**
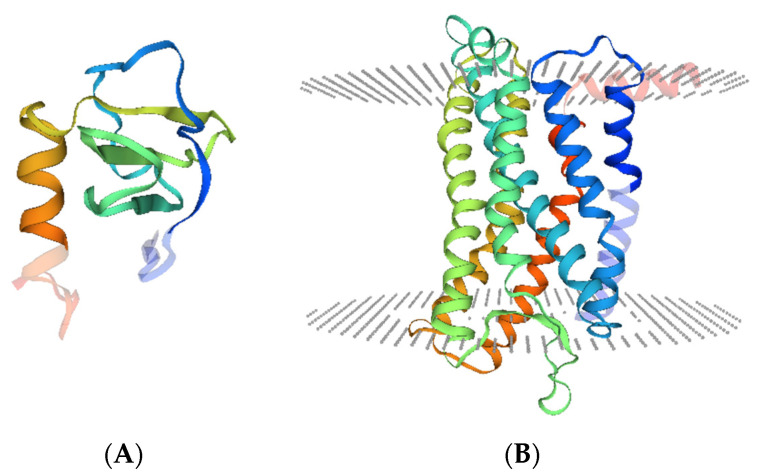
Crystal structure of CXCL13 and CXCR5. (**A**). Illustration of the CXCL13 monomer (UniProKB-O43927) showing domain hits with deep coloration. (**B**). Illustration of the CXCR5 monomer (UniProKB-P32302) showing seven transmembrane helixes and domain hits (deeply colored). Structural models were obtained from SWISS-MODEL (http://swissmodel.expasy.org/repository/. accessed on 15 April 2021).

**Figure 2 life-11-01282-f002:**
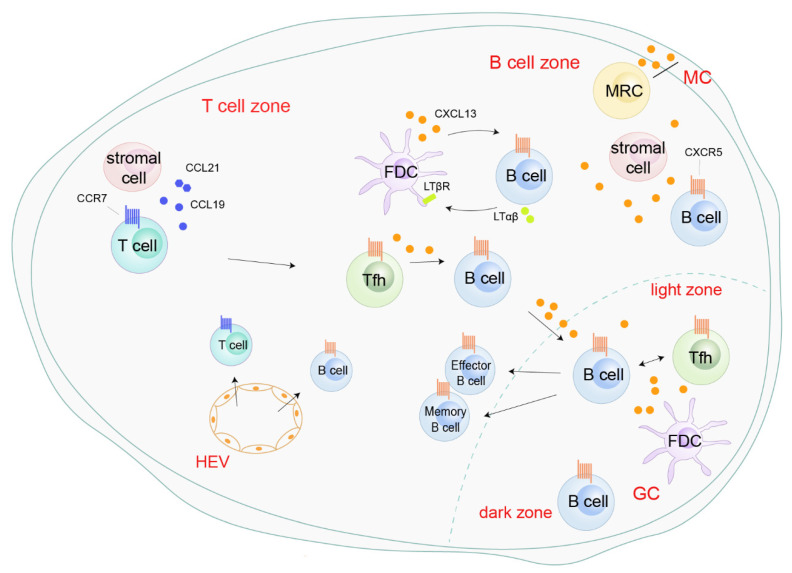
Roles of the CXCL13/CXCR5 axis in secondary lymphoid organs. CXCL13, secreted by follicular helper T cells (Tfh), follicular dendritic cells (FDCs), and marginal reticular cells (MRC), recruits peripheral CXCR5^+^ B cells into the B cell zone or germinal center (GC) through high endothelial venules (HEVs). In the B cell zone, CXCL13 enhances follicle development and sustains secondary lymphoid organ homeostasis by a positive-feedback loop with B cells and FDC [26]. CXCR5^+^ Tfh migrate into B cell zones, initiating GC formation and B cell receptor (BCR) affinity maturation and promoting the differentiation of B cells into antibody-producing plasma cells and memory cells [40,41,42].

**Figure 3 life-11-01282-f003:**
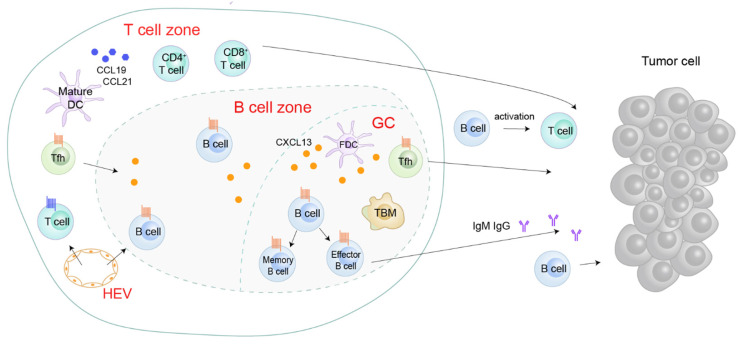
Roles of the CXCL13/CXCR5 axis in the tertiary lymphoid structure (TLS). CXCL13 is aberrantly expressed in the TLS. CXCL13 attracts CXCR5^+^ Tfh and B cells to the B cell zone or GC, potentiating B cell maturation and TLS formation [47,48]. In TLS, B cells’ secret immunoglobulins activate T cells or directly target cancer cells [15,49]. Tfh, follicular helper T cells; FDC, follicular dendritic cells; GC, germinal center; HEV, high endothelial venules; TBM, tingible body macrophages.

**Figure 4 life-11-01282-f004:**
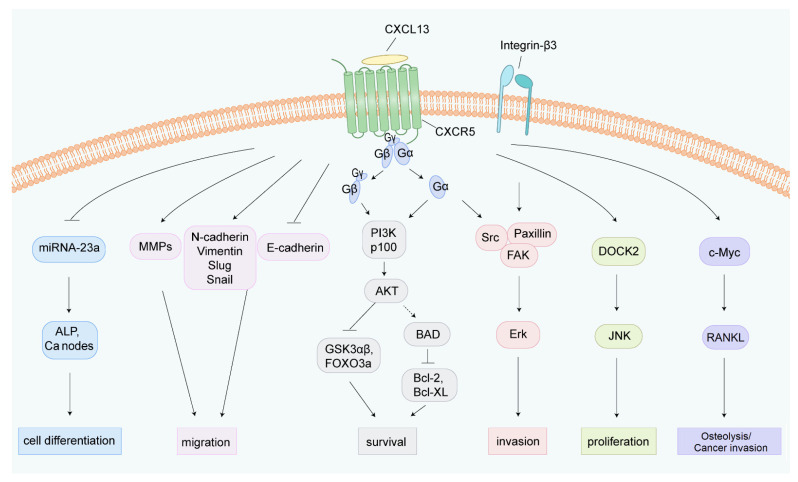
Illustration of the underlying mechanisms of the CXCL13/CXCR5 axis in cell fate determination. The CXCL13/CXCR5 axis triggers multiple intracellular signal transduction pathways. After CXCL13 binds to CXCR5, G proteins dissociate from CXCR5, dividing into Gα and Gβγ, thereby inducing different downstream molecular events [20,23]. CXCL13 promotes osteogenic differentiation by inhibiting miRNA-23a, inducing ALP activity, and calcium node formation [82]. Upregulation of MMPs, N-cadherin, Vimentin, Slug, and Snail, and downregulation of E-cadherin under CXCL13 treatment enhances tumor cell migration [18,83]. The CXCL13/CXCR5 axis activates PI3K/Akt, integrin-β3/Src/Paxillin/FAK, and the DOCK/JNK pathway to induce cell survival, invasion, and proliferation, respectively [73]. CXCL13 increases the phosphorylation of c-Myc and c-Jun, and upregulates the transcriptional regulator NFATc3, which binds to the promoter region of RANKL and elevates the expression of RANKL [74,84]. ALP, alkaline phosphatase; Ca, calcium; MMPs: matrix metalloproteinase; BAD, Bcl-2 agonist of cell death; FAK, focal adhesion kinase; DOCK2, dedicator of cytokinesis 2; JNK, c-Jun kinase; RANKL, receptor activator of NF-kB ligand.

**Figure 5 life-11-01282-f005:**
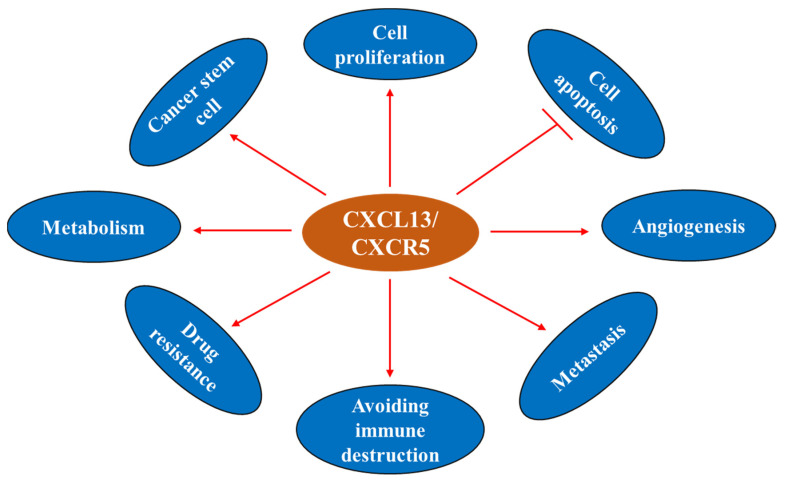
CXCL13 and cancer hallmarks.

**Figure 6 life-11-01282-f006:**
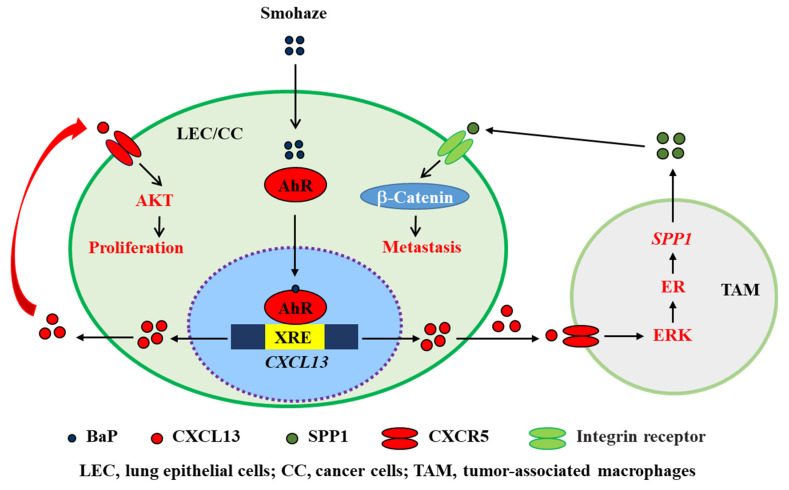
Schematic representation of smohaze-induced production of CXCL13 in lung cancer.

**Figure 7 life-11-01282-f007:**
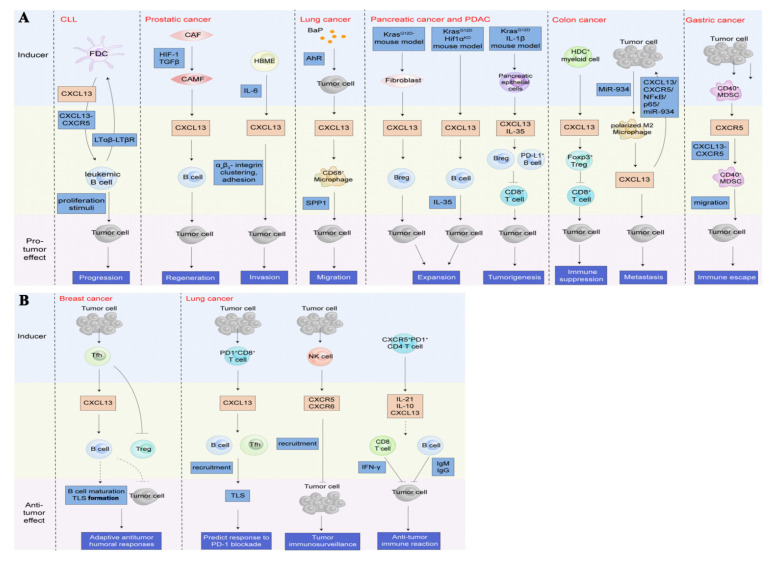
Roles of the CXCL13/CXCR5 axis in the tumor microenvironment (TME). (**A**). The CXCL13/CXCR5 axis plays a crucial role in shaping a complex TME by recruiting multiple types of lymphocytes to beget pro-tumor or anti-tumor immunity reactions. In the context of pro-neoplastic reactions, the CXCL13/CXCR5 axis attracts B cells to induce tumor progression, regeneration, and invasion [86,87,115,118], or recruits CD68^+^ macrophages [18], Breg [117,118], Treg [119], and CD40^+^ MDSC [120] to trigger migration, expansion and tumorigenesis, immune suppression, and immune escape, respectively. (**B**). In the circumstance of anti-tumor reactions, the CXC13/CXCR5 axis recruits B cells and Tfh [89,90] to format TLS in the peritumoral or tumor sites, which is associated with adaptive anti-tumor humoral responses and predicting the response to PD-1 blockade therapy. Additionally, lymphocytes directly or indirectly dampen tumors by the upregulation of CXCL13 and/or CXCR5 [15,121]. CLL, chronic lymphocytic leukemia; LTαβ, lymphotoxin α1β2; LTβR, lymphotoxin-β receptor; CAF, cancer-associated fibroblasts; CAMF, cancer-associated myofibroblasts; HIF-1, hypoxia-inducible factor 1; TGFβ, transforming growth factor-β; HBME, human bone marrow endothelial; IL, interleukin; BaP, benzo(a)pyrene; AhR, Aryl hydrocarbon receptor; SPP1, secreted phosphoprotein 1; PDAC, pancreatic ductal adenocarcinoma; Breg, regulatory B cells; HDC, histidine decarboxylase; Treg, regulatory T cells; MDSC, myeloid-derived suppressor cells; Tfh, follicular helper T cells; NK cell, natural killer cell.

**Table 1 life-11-01282-t001:** Therapeutic targets associated with the CXCL13/CXCR5 axis in malignancies and the tumor microenvironment.

Target	Cancer Type	Function	Approach	In Vivo or In Vitro	Outcome	Refs.
CXCL13	Prostate cancer	Induction of prostate cancer cell proliferation and migration	siRNA and shRNA; antibody	In vivo; in vitro	Inhibiting tumor growth and metastasis	[122]
CXCL13	Prostate cancer	Chemotaxis B cells into regressing tumor	Antibody	In vivo	Preventing B-cell recruitment into tumor under castration	[115]
CXCL13	Breast cancer	Activating CXCR5/ERK pathway	Polyclonal antibody	In vivo; in vitro	Attenuating tumor volume and growth; inhibiting tumor cell proliferation and promoting its apoptosis	[102,104]
CXCL13	Breast cancer	Enhancing the production of RANKL on tumor cells and the interaction between ILC3 and stromal cells	Antibody	In vivo	Attenuating lymph node metastasis	[131]
CXCL13	Lung cancer	Promotion of cell proliferation; inducing the production of SPP1 by microphage	*Cxcl13*^−/−^ mice	In vivo	Decreasing the volume of BaP-induced tumor	[18]
CXCL13	PDAC	Homing B cell into tumor lesions	Antibody	Mice harbored Kras^G12D^ PDEC	Reducing the growth of orthotopic tumor	[116]
CXCL13	Colon cancer	Induction 5-Fu resistance and association with a worse outcome	siRNA	In vitro	Reducing 5-Fu resistance	[112]
CXCR5	CLL	CXCR5^+^ leukemia B cells recruited by CXCL13 to encounter proliferation stimuli	*Cxcr5*^−/−^ *Eμ-Tcl1* mice	In vivo	Attenuating tumor cell proliferation	[86]
CXCR5	Prostate cancer	Induction of prostate cancer cells proliferation and migration	siRNA and shRNA	In vivo; in vitro	Inhibiting tumor growth and metastasis	[122]
CXCR5	Lung cancer	CXCR5^+^ CD68^+^ macrophages producing SPP1 to promote EMT process	*Cxcr5*^−/−^ mice	In vivo	Decreasing the volume of BaP-induced tumor	[18]
CXCR5	OSCC	Induction RANKL expression under CXCL13/CXCR5 axis	Antibody	In vitro	Inhibiting the expression of RANKL	[74]
TGFβR	Prostate cancer	Activating CXCL13-expressing myofibroblasts	SB-431542	In vivo	Blocking the initiation of castration-resistant prostate cancer	[87]
NFATc3	OSCC	Nuclear translocation mediated by CXCL13/CXCR5 axis to bind to RANKL promoter region	siRNA	In vitro	Preventing RANKL expression	[74]
Myofibroblasts	Prostate cancer	Induction of CXCL13 expression	Immunodepletion; phosphodiesterase 5	In vivo	Blocking the initiation of castration-resistant prostate cancer	[87]

ILC3, RORγt^+^ innate lymphoid cell group 3; SPP1, secreted phosphoprotein 1; BaP, benzo(a)pyrene; PDAC, pancreatic ductal adenocarcinoma; PDEC, pancreatic ductal epithelial cells; 5-Fu, 5-Fluorouracil; CLL, chronic lymphocytic leukemia; EMT, epithelial to mesenchymal transition; OSCC, oral squamous cell carcinomas; RANKL, RANK ligand; TGFβR, TGFβ receptor; NFATc3, nuclear factor of activated T cells.

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
