# Peer review of "CXCL13 in Cancer and Other Diseases: Biological Functions, Clinical Significance, and Therapeutic Opportunities"

_life, 2021, doi:10.3390/life11121282_

Round 1

Reviewer 1 Report

Thank you, authors, for the opportunity to review your manuscript. The manuscript is well constructed and encompasses important information about the CXCl13/CXCR5 axis in cancer. There are some suggestions I have that might improve the manuscript quality, they are:

  1. Do benzo(a)pyrene and azoxymethane work in a similar manner as the suggestion comes out in the abstract?
  2. What is the difference between the nuclear localization/cell membrane-localized CXCR5?
  3. I am a little concerned about the duplication of information in the manuscript between sections 4 and 5. 
  4. In my opinion section 4.2 should come before section 4.1. Moreover, the language seems to be replicating the research study found from the cited study.
  5. More input/thoughts/interpretation of the literature by the authors will help.
  6. A more generalized figure will be better for figure 5 rather than a representative figure of each section will be more informative.

Author Response

Reviewer 1:

  1. Do benzo(a)pyrene and azoxymethane work in a similar manner as the suggestion comes out in the abstract?

- We thank the reviewer for the comment. Previous studies showed that BaP undergoes metabolic activation in vivo, resulting in production of BaP-7,8-diol-9,10-epoxides (BPDEs). BPDE reacts with DNA to form adducts at N2 of deoxyguanosine (BPDE-N2-deoxyguanosine), which induces mainly G→T genomic mutations to promote carcinogenesis. Azoxymethane is catalysed into methylazoxymethanol, which induces G→A mutations. Therefore, these carcinogens work in a somewhat similar (but also quite different) manner to induce cancer. We found that deficiency in CXCL13 inhibits both BaP- and azoxymethane-induced carcinogenesis, indicating the critical role of this chemokine in environmental carcinogenesis. The information has been added to the text (page 2 and 3). Thank you.

  1. What is the difference between the nuclear localization/cell membrane-localized CXCR5?

- We thank the reviewer for the interesting question. Studies showed that in non-small cell lung cancer, the expression of CXCR5 in the nuclear is higher than that in the cell membrane (Int J Oncol. 2014;45:2232-40), and CXCR5 neddylation is involved in targeting the receptor to the cell membrane (J Cell Sci 2014;127:3546-54). However, the difference between the nuclear localized and cell membrane-localized CXCR5 remains unclear. One possibility could be that the membranous CXCR5 is modified (i.e., phosphorylation, acetylation, or other types of modification) and translocated into nucleus, to exert novel biological function. This represents an open and interesting question to be answered in the future. The information has been added and the references have been cited (pages 4 and 20).

  1. I am a little concerned about the duplication of information in the manuscript between sections 4 and 5.

- We thank the reviewer for pointing out the potential duplication, and re-organized the content to reduce duplication.

  1. In my opinion section 4.2 should come before section 4.1. Moreover, the language seems to be replicating the research study found from the cited study.

- We thank the reviewer for the suggestion, and revised the manuscript accordingly. We also used our own language to cite the research study.

  1. More input/thoughts/interpretation of the literature by the authors will help.

- More thoughtful information has been included. Thank you.

  1. A more generalized figure will be better for figure 5 rather than a representative figure of each section will be more informative.

- We thank the reviewer for the comment. The original Figure 3 (now Figure 4) and original Figure 4 (now Figure 3) are for general functions of CXCL13/CXCR5 axis in cell fate determination and in cancers. Due to the heterozygosity of cancers and different functions of CXCL13/CXCR5 in different subtypes of cancer, the original Figure 5 (now Figure 6) is not modified. Thank you.

Reviewer 2 Report

Review in CXCL13 in cancers: the sources, its effects on target cells, and 2 therapeutic opportunities

Authors: San-Hui Gao, Sheng-Zhi Liu, Gui-Zhen Wang and Guang-Biao Zhou

General comment:

The authors describe in detail CXCL13 functions, interactions and its role in several types of cancers. The secretion of CXCL13 by several cell types of the tumor microenvironment is also very interesting and justifies the broad role of CXCL13 in the several steps of cancer progression.

Although this is not my area of expertise, I found this paper very interesting, the images well organized and presented. This extensive review will certainly contribute to the progress in the cancer field.

Major comments

  1. No major comments

Minor comments:

  1. English writing should be improved
  2. Line 53: structural should be replaced by structure

Author Response

Reviewer 2: 

Minor comments:

  1. English writing should be improved

- The manuscript has been edited and English writing has been improved. Thank you.

  1. Line 53: structural should be replaced by structure

- “Structural” in line 53 has been replaced by structure. Thank you.

Reviewer 3 Report

Ref: Life-1349187
Title: CXCL13 in cancers: the sources, its effects on target cells, and therapeutic opportunities
Journal: Life-MDPI

The manuscript entitled: “CXCL13 in cancers: the sources, its effects on target cells, and therapeutic opportunities” by Gao et al. is a review article that summarizes the molecular events and TME alterations caused by CXCL13/CXCR5 and discuss the potential of agents targeting this axis in different malignant tumors. Although this manuscript fits within the scope of the journal, it needs major revisions before being considered for publication in the journal “Life”. Please find below the comments-suggestions and recommended revisions:

Major revisions:

-Title: could be more general; for instance: CXCL13 in cancer: translational aspects and therapeutic opportunities or something similar to reflect better the content

-Abstract: line 18-19: “in most subtypes of cancers”: should be rephrased to “in most cancer types”. This is also a general comment; please make this correction throughout the text.

-CXCL13/CXCR5 and benign diseases: a benign disease usually is mentioned for a tumor that is not malignant. To avoid confusing would be better to replace the term “benign diseases” to “other diseases” since you mention different types of diseases. Moreover, Figure 2B shows the impact on tumor cells; however, this section refers to other types of diseases, other than cancer. The figure placement does not reflect the content here.

-Figure 3: this figure is very simple; you should include more details (i.e showing where CXCL13/CXCR5 is involved and in which mechanisms in the hallmarks of cancer).

-Figure 4: Please have a look on the cMYC—NFkB---Gene transcription---where does this lead, activation of which genes? Also please include this figure within the text.

-Figure 5:  the description of the figure belongs to the TME; however you included this figure in the section “the sources of CXCL13 in tumors”; please elaborate.

- The sources of CXCL13 in tumors: it would be better to be described as CXCL13 sources within the tumor and the tumor microenvironment -Also the part of the paragraph (lines 363-373) should be re-written better to reflect the tile of this section.

-General comment: In general there is so much important information in the text that is not shown in the text; you should at least include a few more major points in the figures or produce another figure.

-Line 632-633: “High infiltration of CXCL13+ CD8+ T cells in tumor tissue link to poor clinical outcome and could be a potential target for gastric cancer immunotherapy”: please have a look on this statement and explain better.

- 6. Therapeutic potentials of CXCL13/CXCR5 axis in cancer: authors focus mainly on the CXCL13/CXCR5 signaling in B cells; however you should expand and include the information given in table 1 in this section and also mention the table in the text. This part needs to be re-written. Also, you should expand and explain the implication on the other therapies apart from immunotherapy.

-General comment: more recent references could have been included in this review article.

Minor revisions:

-Figure 6: the figure legend is very brief; you should include and describe the figure briefly here.

-Figure 1B: please replace the word “cartoon” with another work (perhaps illustration).

-General comment: Please check all the grammatical and typo errors within the whole manuscript.

-Table 1: could also be mentioned in section 5. CXCL13/CXCR5 in several subtypes of cancers.

- 5.6. Coloreatal cancer: please correct as “colorectal”

-Line 642: Please replace the “therapeutic potentials” to “therapeutic potential”.

- Line 662: “CXCL13/CXCR5-producing nonneoplastic cells”: please mention which are those cells.

-Table 1: Please correct the SiRNA to siRNA. Also, please have a look on the first column. Maybe would be better to reconsider including the last raw in this table (Myofibroblasts), since does not fit to the rest of the content.

Author Response

Reviewer 3:

Major revisions:

-Title: could be more general; for instance: CXCL13 in cancer: translational aspects and therapeutic opportunities or something similar to reflect better the content

- Thank you for the comments. The title has been changed to “C-X-C chemokine ligand 13 in immune homeostasis and diseases: biological functions, clinical significance, and therapeutic opportunities”.

-Abstract: line 18-19: “in most subtypes of cancers”: should be rephrased to “in most cancer types”. This is also a general comment; please make this correction throughout the text.

- As suggested, we revised the sentences accordingly. Thank you.

-CXCL13/CXCR5 and benign diseases: a benign disease usually is mentioned for a tumor that is not malignant. To avoid confusing would be better to replace the term “benign diseases” to “other diseases” since you mention different types of diseases. Moreover, Figure 2B shows the impact on tumor cells; however, this section refers to other types of diseases, other than cancer. The figure placement does not reflect the content here.

- The term “benign diseases” has been replaced by “non-cancerous diseases”. The original Figure 2B has been removed from Figure 2 and used as an independent figure (Figure 5 in the revised manuscript). Thank you for the important comments.

-Figure 3: this figure is very simple; you should include more details (i.e showing where CXCL13/CXCR5 is involved and in which mechanisms in the hallmarks of cancer).

- We appreciate your advice. Yes the original Figure 3 (now Figure 5) is simple, but many molecules participate in regulating cancer hallmarks. If these molecules/pathways are included, the figure will be very complicated and busy. So it would be better to keep the figure simple and neat? Thank you for the constructive suggestion and understanding. 

-Figure 4: Please have a look on the cMYC—NFkB---Gene transcription---where does this lead, activation of which genes? Also please include this figure within the text.

- Thank you for the comments. We revised Figure 4, which now shows that CXCL13 regulates c-Myc-RANKL-osteolysis/cancer invasion. Thank you.

-Figure 5:  the description of the figure belongs to the TME; however you included this figure in the section “the sources of CXCL13 in tumors”; please elaborate.

- We thank the reviewer for pointing out this. In the revised text, we included the original Figure 5 (now Figure 7) in the section “CXCL13/CXCR5 in tumor microenvironment” (section 4.2.5) and “CXCL13/CXCR5 in several cancer types”. Thank you.

- The sources of CXCL13 in tumors: it would be better to be described as CXCL13 sources within the tumor and the tumor microenvironment -Also the part of the paragraph (lines 363-373) should be re-written better to reflect the tile of this section.

- Thank you for the important comments and suggestions. We renamed “The sources of CXCL13 in tumors” to “CXCL13 sources within the tumor and the tumor microenvironment” and move this section from 4.2 to 4.1.

The part of the paragraph (lines 363-373) has been revised: the tile of this section has been changed to “5.1 Chronic lymphocytic leukemia and lymphoid neoplasms”, and this section comprises “5.1.1 Chronic lymphocytic leukemia” and “5.1.2 Lymphoid neoplasms”. Thank you.

-General comment: In general there is so much important information in the text that is not shown in the text; you should at least include a few more major points in the figures or produce another figure.

- We thank the reviewer for the comment. The manuscript now has 7 figures and 1 table, most of the important information has been included in the figures and table. Though there is still important information in the text that is not shown in the figures or the table, we were unable to produce another figure this time. Thank you.

-Line 632-633: “High infiltration of CXCL13+ CD8+ T cells in tumor tissue link to poor clinical outcome and could be a potential target for gastric cancer immunotherapy: please have a look on this statement and explain better.

- The sentence has been replaced by “High infiltration of CXCL13+ CD8+ T cells in tumor tissue is associated with poor clinical outcome of the patients, and elimination of these cells could be helpful for gastric cancer immunotherapy”. Thank you.

- 6. Therapeutic potentials of CXCL13/CXCR5 axis in cancer: authors focus mainly on the CXCL13/CXCR5 signaling in B cells; however you should expand and include the information given in table 1 in this section and also mention the table in the text. This part needs to be re-written. Also, you should expand and explain the implication on the other therapies apart from immunotherapy.

- We thank the reviewer for the important comment, and revised the manuscript accordingly. Table 1 has been cited in the text, and the implication on the other therapies apart from immunotherapy has been expanded. Thank you.

-General comment: more recent references could have been included in this review article.

- As suggested, we added more recent references (e.g., Ref 19, 49-51, 68, 86, and others) to the article.

Minor revisions:

-Figure 6: the figure legend is very brief; you should include and describe the figure briefly here.

- Thank you for the suggestion. We added more description for the legend of Figure 6.

-Figure 1B: please replace the word “cartoon” with another work (perhaps illustration).

- As suggested, we replaced the word “cartoon diagram” with “Illustration”.

-General comment: Please check all the grammatical and typo errors within the whole manuscript.

- As suggested, we checked and revised the grammar and typo errors. Thank you.

-Table 1: could also be mentioned in section 5. CXCL13/CXCR5 in several subtypes of cancers.

- As suggested, we also labeled the “table 1” in the section 5.

- 5.6. Coloreatal cancer: please correct as “colorectal”

- Thank you for the comment. The word has been corrected.

-Line 642: Please replace the “therapeutic potentials” to “therapeutic potential”.

- As suggested, we revised the word.

- Line 662: “CXCL13/CXCR5-producing nonneoplastic cells”: please mention which are those cells.

- Thank you very much for your advice. We added the detail of nonneoplastic cells.

-Table 1: Please correct the SiRNA to siRNA. Also, please have a look on the first column. Maybe would be better to reconsider including the last raw in this table (Myofibroblasts), since does not fit to the rest of the content.

- Thank you for the suggestion. We corrected the SiRNA to siRNA

Round 2

Reviewer 3 Report

Ref: life-1349187
Title: CXCL13 in cancers: the sources, its effects on target cells, and therapeutic opportunities
Journal: Life-MDPI

The manuscript entitled: “CXCL13 in cancers: the sources, its effects on target cells, and therapeutic opportunities” by Gao et al., is a review article that summarizes the molecular  events and TME alterations caused by CXCL13/CXCR5 and discuss the potential of agents targeting this axis in different malignant tumors. The authors addressed all the major comments raised and now the manuscript has been improved and can be considered for publication in the journal “Life”.

Just one minor point (optional): The title could be modified as: “CXCL13 in cancer and other diseases: biological functions, clinical significance, and therapeutic opportunities”.